# Influence of Illite and Its Amine Modifications on the Self-Adhesive Properties of Silicone Pressure-Sensitive Adhesives

**DOI:** 10.3390/ma16072879

**Published:** 2023-04-04

**Authors:** Adrian Krzysztof Antosik, Karolina Mozelewska, Marlena Musik, Piotr Miądlicki, Katarzyna Wilpiszewska

**Affiliations:** 1Department of Chemical Organic Technology and Polymeric Materials, Faculty of Chemical Technology and Engineering, West Pomeranian University of Technology in Szczecin, 70-322 Szczecin, Poland; 2Engineering of Catalytic and Sorbent Materials Department, Faculty of Chemical Technology and Engineering, West Pomeranian University of Technology in Szczecin, 70-322 Szczecin, Poland

**Keywords:** amine modification, illite, silicone pressure-sensitive adhesive, self-adhesive properties, adhesion

## Abstract

Obtaining new silicone self-adhesive in the presence of modified illite has been described. The filler was modified with N,N,4-trimethylaniline. The effect of illite content and modification on functional properties (adhesion, cohesion, stickiness, and shrinkage) was determined. Additionally, the thermal resistance (the SAFT test) of obtained silicone pressure-sensitive adhesives was evaluated. For all the systems tested, an increase in thermal resistance and shrinkage decrease were noted. Moreover, only a slight adhesion and tack decrease was revealed. Such self-adhesives could be applied for joining elements operating at increased temperatures, e.g., in heavy industry.

## 1. Introduction

An adhesive is a substance capable of permanently bonding two surfaces as a result of the forces of adhesion to the surface and the internal cohesion of the adhesive joint. Adhesives play an increasingly important role in many industries, from the production of toys to the construction of aircraft [1,2].

The term “pressure-sensitive adhesives” covers adhesives based on high-molecular-weight polymers characterized by significant stickiness (tack), excellent adhesiveness, and high cohesion [3,4].

The self-adhesives are widely used in packaging, advertising, printing, medicine, construction, automotive, and machine industries. They are used as insulating, soundproofing, sealing, and filling material, during assembly, as gaskets, washers, vibration damping and absorbing elements, anti-slip pads, as decorative films, security, and joining materials in places particularly exposed to extreme conditions (resistant to UV radiation, weather conditions, and aggressive chemicals). Such products meet high European standards, and the worldwide guarantee of the desired functions and properties of the products is of particular importance, especially in the case of major application areas, such as the automotive industry [5,6,7].

Due to the adhesion mechanism, the adhesives can be divided into solvent-based, water dispersions, and solvent-free adhesives. Solvent-based adhesives penetrate deeply into the material, causing its swelling and partial dissolution. The glued surfaces are pressed, and the solvent evaporates, leaving a permanent joint. Silicone adhesives are an example of solvent-based adhesives. Thanks to their properties, they are universal products that are appreciated in many areas. They are distinguished by excellent flexibility and excellent adhesion to various substrate types. They bind very quickly and permanently to painted surfaces, ceramics, glass, and aluminum. Additionally, silicone adhesives guarantee stability in a wide temperature range, which widens their applications’ range. Such joints are resistant to various weather conditions (including UV radiation), chemically stable (resistant to many different chemicals and pollutants in the air) and have low flammability. Due to their electrical properties, they can be used in various electrical and electronic applications, including devices exposed to significant temperature fluctuations [4,8,9].

The base component that gives the adhesive properties is always the polymeric substance. However, the other components may also be solvents, fluidizing agents, thinners, fillers, accelerators, wetting agents, or stabilizers. Generally, the additives and fillers usually modify the individual properties of the adhesive, such as, for example, viscosity, wettability, adhesion strength, cohesive strength, and heat resistance color, but do not change the basic properties of the adhesive [7,10,11].

Illite is a natural clay mineral that is a member of the 2:1 family of layered silicates. Its crystal structure comprises two silicon-oxygen tetrahedrons fused to an edge-shared aluminum−oxygen octahedron. Illite exhibits moderate cation exchange capacity, large specific surface area, and low production cost; currently, the world has huge reserves [12]. The distinction between the different types of illite lies mainly in the location of the mining location. To improve the compatibility between hydrophilic zeolite filler and hydrophobic polymer matrix, the mineral must be modified. Recently, the amine modification of zeolites has received scientific attention. Sodium montmorillonite was modified with linear amines and oligomeric oxypropylene amine derivatives by cation exchange at room temperature [13].

Raw illite was intercalated with cetyltrimethylammonium bromide or polar hydrazine hydrate (N_2_H_4_). First, the natural illite was suspended with deionized water using an ultrasonic technique. Then, N_2_H_4_ was added, and the mixture was agitated for 4 h at 40 °C before being ultrasonically processed for 1 h. The product was ground and dried at 70 °C under a vacuum. The modification using cetyltrimethylammonium bromide was reported. An aqueous suspension of hydrochloric acid and cetyltrimethylammonium bromide was added to illite. The mixture was stirred at 80 °C for two hours and then let to settle for 24 h. Subsequently, it was dried at 70 °C under a vacuum and ground. The modified illite was used as a filler for polypropylene composites. The illite hydrazine hydrate modification methods presented as an example are quoted from the literature [14].

The aim of this work was to develop new silicone-based self-adhesives exhibiting improved thermal resistance. The adhesive was used to obtain one-side pressure-sensitive self-adhesive tape. The mineral filler—illite, modified (etched) with N,N,4-trimethylaniline, was used. The effect of illite modification and content on viscosity, peel adhesion, cohesion (at 20 and 70 °C), tack, shrinkage, and heat resistance (SAFT test) were evaluated. Additionally, the pot life of obtained adhesive was evaluated. The tapes with improved thermal resistance going beyond the currently performed stability in a wide range of temperatures could be applied for joining elements operating at increased temperatures, e.g., in heavy industry.

## 2. Materials and Methods

### 2.1. Materials

The silicone resin DOWSIL™ 7358 (Q2-7358) was purchased from Dow Corning (Midland, MI, USA). The crosslinking agent, dichlorobenzoyl peroxide (DClBPO), was a product of Gelest (Bucks County, PA, USA). Illite was purchased from Nanga (Blękwit, Poland), toluene from Carl Roth (Karlsruhe, Germany), and N,N,4-trimethylaniline from Merck (Darmstadt, Germany).

### 2.2. Filler Modification

Illite was modified with N,N,4-trimethylaniline (THA). The illite suspension 10 g in 100 mL THA solution (various concentrations were applied: 0.1 M, 0.5 M, and 1 M) was ultrasonically treated for 6 h at 60 °C in an ultrasonic bath to etch the organic part of the filler. The modified illite was filtered and washed with distilled water. The samples were dried at 100 °C for 24 h. The symbols of modified fillers are presented in Table 1.

### 2.3. Preparation of One-Side Self-Adhesive Tape

Silicone resin Q2-7358 was mixed with modified illite and a crosslinker (1.5 wt.% 2-4-dichlorobenzoyl peroxide). The tested filler content was 0.1, 0.5, 1, and 3 wt.% (based on dry polymer). The prepared composition was coated (45 g/m^2^) onto a polyester film (50 g/m^2^) using a semi-automatic coater. The adhesive film was then placed in a drying duct for 10 min at 110 °C for crosslinking. Subsequently, the obtained adhesive film was secured with a second layer of fluorosiliconized polyester film (produced by Dolpap, Chojnów, Poland) and cut into strips for further tests.

### 2.4. Infrared Spectroscopy FTIR

The Nicolet 380 spectrometer (Thermo Electron Corporation, Waltham, MA, USA) with ATR attachment (diamond crystal) was used. The measurements were performed with a resolution of 0.4 cm^−1^ and a range of 4000–400 cm^−1^.

### 2.5. X-ray Diffraction (XRD)

To investigate the modified and pristine illite structure, the Empyrean PANalytical X-ray diffractometer (Malvern, UK) was used. The radiation source was a Cu lamp, the 2θ was in a range of 10–70°, and the step size was 0.026.

### 2.6. Pot-Life

The pot life is the maximum time after preparation when the composition can be coated on a substrate. Practically, it refers to the time required to increase the viscosity (twice or four times) compared to the initial viscosity value. The test is carried out at room temperature and starts immediately after preparing the composition. Tests were carried out on 2 samples for each formulation to determine the mean pot life, and the standard deviation was used as the error.

### 2.7. Adhesion

Adhesion is defined as the interaction between two surfaces. It closely relates to the interfacial attraction forces and is one of the most important bonding phenomena. The work test was performed according to the standard Fédération Internationale des Fabricants et Transformateurs d’adhesifs et Thermocollants sur Papiers et Autres support (FINAT) FTM 1—Peel adhesion (180°) a 300 mm per minute. Tests were carried out on 6 samples for each formulation to determine the mean adhesion, and the standard deviation was used as the error.

### 2.8. Cohesion

Cohesion refers to the strength of the adhesive joint. Besides adhesion, it is one of the most crucial properties of adhesives. Its value depends on the test temperature, type, and content of the crosslinking agent or the thickness of the adhesive film. It can be determined using the standard FINAT FTM 8—Resistance of shear from a standard surface. The measurement was carried out at the room and elevated temperature (70 °C). Tests were carried out on 4 samples for each formulation to determine the mean cohesion at room and elevated temperature, and the standard deviation was used as the error.

### 2.9. Tack

Tack refers to the ability of an adhesive to bond without pressure with the other surface. It can also be defined as the force needed to separate the surfaces after a short time. It was measured according to the technical standard FINAT FTM 9—“Loop” tack measurement. Tests were carried out on 6 samples for each formulation to determine the mean tack, and the standard deviation was used as the error.

### 2.10. Thermal Resistance

The thermal resistance was evaluated using the SAFT test (Shear Adhesion Failure Temperature). A 1 kg weight was hung on the lower end of the one-sided adhesive tape and placed in the oven. The temperature was increased from 22 °C to 225 °C at a heating rate of 1 °C/min. Each time, the damage temperature (the temperature at which the tape detached from the plate) was recorded by reading in the computer, and then the nature of the adhesive damage—adhesive or cohesive cracking—was checked organoleptically. Tests were carried out on 4 samples for each formulation to determine the mean thermal resistance, and the standard deviation was used as the error.

### 2.11. Shrinkage

The shrinkage was tested by the standard FINAT FTM 14—Dimensional stability. The PVC foil (10 × 10 cm^2^) with a self-adhesive adhesive was placed onto an aluminum plate. In the center of the sample, a vertical and horizontal incision was made to form a cross (incisions with a dimension of 8 cm). Next, the sample was seasoned at 70 °C. The shrinkage was determined after 10 min, 0.5, 1, 3, 8, 24 h, and 2, 3, 4, 5, 6, and 7 days (width of the slits formed by the cuts).

### 2.12. Thermogravimetric Analysis

TGA was carried out using thermomicrobalance from NETZSCH (Selb, Germany) with a scan range from 25 °C to 800 °C at a constant heating rate of 10 °C/min in an air atmosphere with nitrogen flow as the purge gas. Samples of 9–10 mg were loaded in an Al_2_O_3_ crucible.3. Results and discussions

Table 2 shows the corresponding IR band assignments of illite and modified illite (Figure 1). The characteristic bands of 3697, 3625, and 3432 cm^−1^ correspond to the stretching vibration of the hydroxyl groups (OH). The bands at 1033 and 468 cm^−1^ are those of Si-O bonds. The band located at 911 cm^−1^ is attributed to the deformation vibration of Al-OH, and the bands at 753 and 532 cm^−1^ arise from the deformation vibrations of Al-O-Si. The presence of a doublet at 789 and 777 cm^−1^ and a singlet located at 693 cm^−1^ is attributed to the quartz vibrations. The band at 1455 cm^−1^ developed due to the hydroxyl bending vibration again reflects the presence of bound water [15,16].

As shown in the XRD pattern of Figure 2, the dominant phase is quartz (JCPDS card No. 46–1045), an impurity present in the illite sample, where on the figure Q and I mean respectively quartz and illite. The peaks at 2θ = 17.77°, 19.90°, 27.99°, 26,65 and 35.05°, according to JCPDS card No. 000-26-0911 corresponds to the illite phase. Natural clay, not fractionated, contains a great deal of quartz (the main impurity), and this is a normal phenomenon. In addition, the intensity of the XRD peaks varies depending on the substance being determined and its crystallinity, so peak intensities should not be considered without appropriate calibration. Amine modification has no effect on the crystalline structure of the materials presented; only amorphous impurities may have been removed during the modification—the amine etching. This confirms the identity of the IR spectra of the pure filler and its modified versions. During the modification of the etching, the amines did not graft onto the surface of the filler.

All samples showed similar patterns with an almost continuous and significant mass loss from 25–700 °C (Figure 3), a combination of water and amine loss resulting from modification. The mass loss increases with the amine concentration, indicating that higher concentrations have increased the amount of amine deposited on the material. The highest loss was observed for the THA 1.0 sample, while the unmodified sample exhibited the lowest.

The base silicone pressure-sensitive adhesive without the filler exhibited good adhesion and tack properties—presented in Table 3. The cohesion at room temperature and 70 °C was also high. However, the thermal resistance of pressure-sensitive adhesives revealed a relatively low SAFT test result. This, unfortunately, limits the applicability of the adhesive. The goal of the filler addition and its modification was to improve the thermal resistance of the silicone adhesive while maintaining other parameters on a similar level.

Table 4 presents the viscosity changes over time of the adhesives containing 3 wt.% illite. The viscosity of the compositions increased rapidly after filler addition (i.e., the viscosity doubled the value or was too high to be coated on a substrate) when compared to the neat composition. The lowest value was obtained for the sample containing filler without modification. The highest viscosity increase was noted between the fifth and the seventh day. Practically all the systems, after 7 days, became too viscous to be coated.

Comparing the fillers used, the filler modified with amine with a concentration of 0.1 M shows the lowest viscosity value, while the highest viscosity value was obtained for the filler modified with amine of 1.0 M. This may be due to the fact that illite filler was etched with amine with the highest concentration; therefore, the proportion of silicon atoms is the highest [17]. The desired viscosity of the modified compositions is as low as possible, depending on the type of material to be coated; the upper limit is assumed to be that the viscosity around 50 Pas is already uncoatable (paste consistency).

Figure 4 shows the peel adhesion values of silicone pressure-sensitive adhesives containing modified illite. The blue color shows the results for the adhesive modified with the filler, while the filler was not treated with amine. The red color shows the results for the tape containing a filler modified with amine at a concentration of 0.1. Green-colored tapes are made of adhesive modified with a filler etched with amine at a concentration of 0.5. Moreover, the color purple—a filler treated with amine with the highest concentration equal to 1.0. The adhesive of the system with unmodified illite decreased with filler content. A similar phenomenon was observed for the adhesives containing illite treated with amine. It is a common phenomenon in the technology observed, for example, in adding commercial silicone pressure-sensitive adhesive montmorillonite and its amine modifications [18]. The increase in adhesion with an increase in the filler concentration for the sample containing the filler modified with amine 0.5 is quite a rare phenomenon that is difficult to interpret. It shows the maximum with a content of 1.0 wt.%, likewise for tapes manufactured using illite filler modified with 1.0 M amine. For these samples, a maximum was obtained at the value of 0.5 wt.% of filler. The adhesion value depends on the compatibility between the filler and the matrix, its miscibility, and its tendency to agglomerate [19,20]. However, it seems that each filler’s modification could have a slightly different effect on adhesion value.

The effect of the amount of filler content on the tack is presented in Figure 5. The adhesive containing neat filler showed a known phenomenon, i.e., tack value decreased with filler content [21]. For the composition with a filler modified with the lowest concentration of THA, a slight increase was observed at the value of 1.0 and 3.0 wt.%. The system containing illite modified with 0.5 M THA tack value changed only slightly with the filler amount. However, the adhesive with a filler was modified with the highest concentration of amine. Thus, depending on its content, the filler had a different effect on the properties of pressure-sensitive adhesives. It was observed, especially for low and high polymer matrix loads [19,22,23]. Such a phenomenon could be due to the compatibility of the filler with the polymer matrix leading to increased tack for low filler contents. On the other hand, the stickiness value could decrease for higher filler loads as a result of its agglomeration.

Table 5 presents the results of the cohesion at room temperature, 70 °C, and the thermal resistance determined using the SAFT test. In the case of cohesion measurements, all tested compositions showed values higher than required for the industrial strip production (above 72 h). The only exception was lower cohesion (32 h) for the system containing 3 wt.% of neat illite. These results confirm the proper compatibility of the silicone resin with the modified filler.

The prepared self-adhesive tapes were also tested for thermal resistance using the SAFT test. When comparing the result for the neat adhesive (without the filler—Table 2), i.e., 147 °C, it could be observed that for each composition containing illite, increased thermal resistance was noted, even above the maximum tested 225 °C. However, the thermal resistance decreased with increasing the filler amount, which was easily observed for the system containing illite modified with 1.0 M THA. Thus, the presence of mineral filler improved the thermal resistance, but high organic modifier content could be unfavorable. The highest values were obtained for the samples with the filler unmodified with amine. In the case of amine-etched samples, a decrease in the maximum operating temperature was noted, but not below the values obtained by the samples without fillers. Therefore, the etching of the samples improves their compatibility with the resin but does not increase its thermal resistance, which may be related to the increase in silicon mass in the etched fillers compared to the fillers without modification [19,23].

The effect of filler content on the shrinkage of silicone pressure-sensitive adhesives is shown in Table 6. It could be noted that the higher the filler content in the polymer matrix, the lower shrinkage. This may be due to a better alignment of the polymer mesh or a more compact internal structure of the adhesive film [24,25]. In addition, adding a filler not modified with amine has an unfavorable effect on the shrinkage of the pressure-sensitive adhesive. This is because the introduced modifying material affects the speed of the crosslinking reaction of the adhesives in the measured samples, which results in the lack of a compact composite structure and, therefore, compared to the samples modified with amine, they showed less shrinkage [26,27]. The best results after stabilization of the composition (7th day), i.e., the lowest shrinkage values, were obtained for the compositions containing neat illite and the filler modified with 0.1 M THA for fillings 1.0 and 3.0 wt.%.

## 3. Conclusions

New silicone self-adhesives containing modified illite modified with N,N,4-trimethylaniline were obtained. To improve the compatibility of the filler with the adhesive resin, illite was modified with N,N,4-trimethylaniline through etching.

The addition of modified and neat illite resulted in a slight adhesion and tack increase, whereas a high level of cohesion was maintained. The systems with the lowest filler content exhibited high thermal resistance reaching the measuring limit of the SAFT test.

The new self-adhesive materials developed based on the presented tests show a higher thermal resistance than the unmodified tapes available on the market while having a slight impact on the performance properties (adhesion, cohesion) recorded for materials not modified with fillers. The one-sided adhesive tapes with improved thermal resistance could be used for joining elements operating at increased temperatures, e.g., in heavy industry.

## Figures and Tables

**Figure 1 materials-16-02879-f001:**
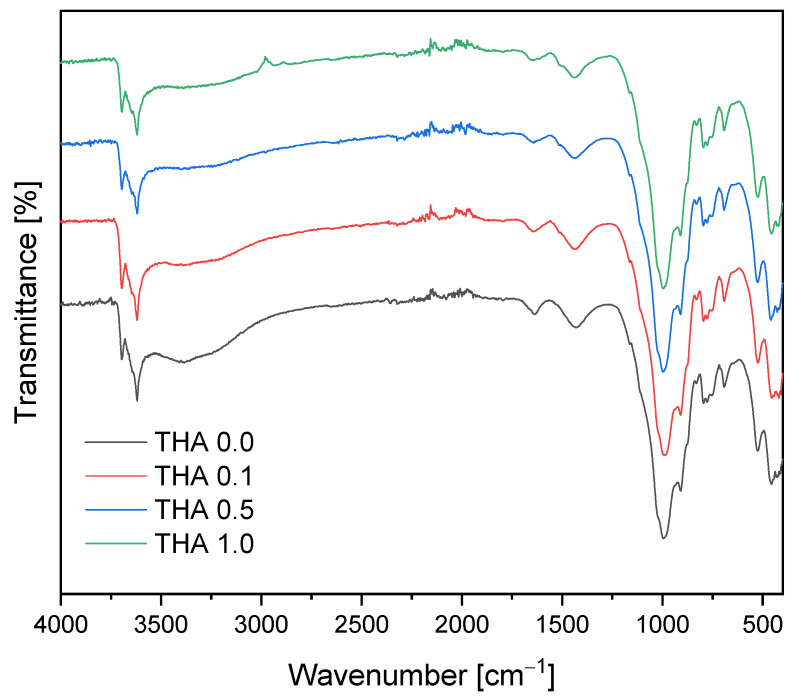
FTIR spectra of raw and modified illite.

**Figure 2 materials-16-02879-f002:**
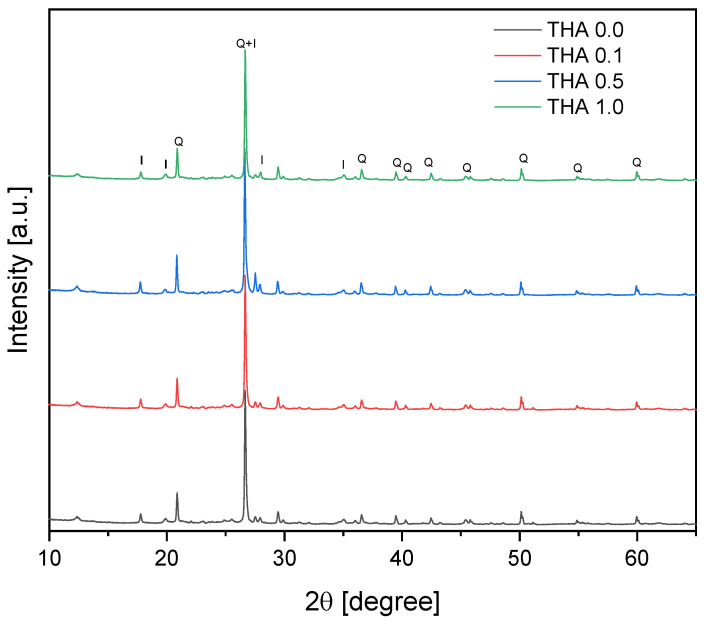
XRD patterns of illite samples.

**Figure 3 materials-16-02879-f003:**
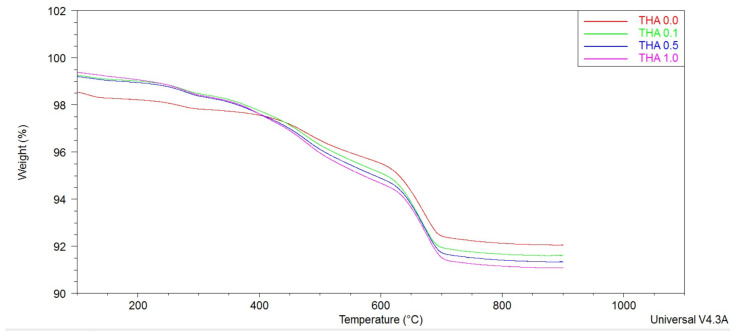
TGA diagrams of pure and modified illite.

**Figure 4 materials-16-02879-f004:**
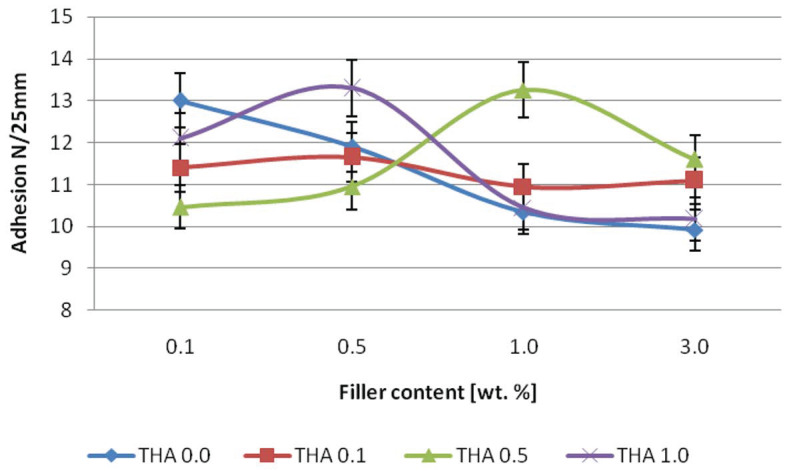
Effect of neat and modified illite content on the peel adhesion of silicone pressure-sensitive adhesive.

**Figure 5 materials-16-02879-f005:**
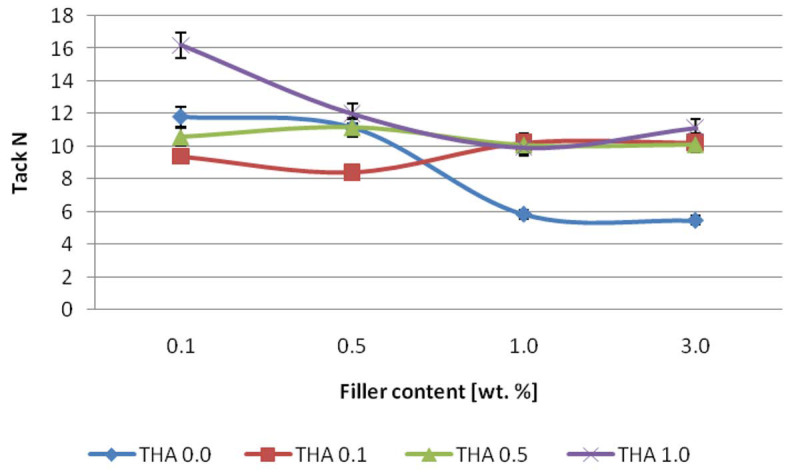
Effect of neat and modified illite on the tack of silicone pressure-sensitive adhesive.

**Table 1 materials-16-02879-t001:** Symbols of natural and modified illite.

Filler	Treatment	Symbols
Illite	Natural, unmodified	THA 0.0
	0.1 M solution of N,N,4-trimethylaniline	THA 0.1
	0.5 M solution of N,N,4-trimethylaniline	THA 0.5
	1.0 M solution of N,N,4-trimethylaniline	THA 1.0

**Table 2 materials-16-02879-t002:** The principal IR bands and their corresponding species of raw and modified illite.

Wavenumber (cm^−1^)	Corresponding Species
3432, 3625, 3697	Hydroxyl groups
1647	H-O-H of water
1455	Hydroxyl groups
1033	Si-O
911	Al-OH
789, 777	Doublet of quartz
693	Si-O-Si of quartz
532, 753	Al-OSi
468	Si-O

**Table 3 materials-16-02879-t003:** Physical properties of base silicon adhesive (without filler).

Resin Acronym	Tack [N]	Adhesion [N/25 mm]	Cohesion [h]		SAFT [°C]	Viscosity [Pa·s]
			20 °C	70 °C		
Q2-7358	6.9	10.2	>72	>72	147	16.7

**Table 4 materials-16-02879-t004:** Viscosity of silicon compositions containing 3 wt.% of modified filler.

Filler	Viscosity [Pa·s]				
	1 Day	2 Days	3 Days	5 Days	7 Days
THA 0.0	21.0	22.8	25.1	28.7	31.5
THA 0.1	39.2	41.0	49.2	58.0	69.0
THA 0.5	35.2	36.9	44.8	53.0	71.0
THA 1.0	33.9	38.5	46.1	54.5	75.0

**Table 5 materials-16-02879-t005:** Cohesion and thermal resistance prepared silicon Q2-7358 composition with different filler content.

Filler Content [wt.%]	THA 0.0	THA 0.1	THA 0.5	THA 1.0
Cohesion at 20 °C [h]				
0.1	>72	>72	>72	>72
0.5	>72	>72	>72	>72
1.0	>72	>72	>72	>72
3.0	>72	>72	>72	>72
Cohesion at 70 °C [h]				
0.1	>72	>72	>72	>72
0.5	>72	>72	>72	>72
1.0	>72	>72	>72	>72
3.0	32	>72	>72	>72
SAFT [°C]				
0.1	>225	>225	>225	>225
0.5	>225	215	>225	206
1.0	>225	213	219	165
3.0	>160	156	180	160

**Table 6 materials-16-02879-t006:** Shrinkage of silicone pressure-sensitive adhesives with different filler content.

Shrinkage [%]												
Filler Content [wt.%]	10 min	30 min	1 h	3 h	8 h	24 h	2 Days	3 Days	4 Days	5 Days	6 Days	7 Days
Pure												
0.0	0.41	0.42	0.64	0.90	0.96	1.02	1.14	1.33	1.33	1.33	1.33	1.33
THA 0.0												
0.1	0.23	0.32	0.35	0.39	0.42	0.47	0.50	0.52	0.55	0.57	0.60	0.61
0.5	0.10	0.12	0.17	0.22	0.28	0.30	0.33	0.36	0.38	0.40	0.40	0.40
1.0	0.08	0.13	0.16	0.19	0.20	0.23	0.25	0.26	0.28	0.29	0.29	0.29
3.0	0.04	0.06	0.08	0.11	0.13	0.14	0.18	0.21	0.23	0.24	0.25	0.20
THA 0.1												
0.1	0.24	0.30	0.33	0.36	0.41	0.46	0.49	0.53	0.55	0.58	0.59	0.60
0.5	0.14	0.17	0.21	0.26	0.31	0.34	0.36	0.39	0.40	0.40	0.40	0.45
1.0	0.09	0.10	0.14	0.16	0.18	0.20	0.23	0.25	0.27	0.30	0.30	0.30
3.0	0.07	0.08	0.10	0.12	0.13	0.14	0.16	0.18	0.20	0.21	0.21	0.21
THA 0.5												
0.1	0.10	0.20	0.28	0.32	0.35	0.38	0.43	0.47	0.49	0.52	0.56	0.58
0.5	0.15	0.18	0.20	0.26	0.30	0.33	0.35	0.38	0.40	0.43	0.48	0.48
1.0	0.10	0.13	0.16	0.20	0.25	0.31	0.33	0.35	0.37	0.40	0.40	0.40
3.0	0.08	0.12	0.14	0.16	0.18	0.20	0.22	0.25	0.27	0.27	0.27	0.25
THA 1.0												
0.1	0.25	0.30	0.35	0.37	0.39	0.43	0.46	0.48	0.51	0.54	0.55	0.55
0.5	0.21	0.25	0.28	0.31	0.35	0.38	0.40	0.43	0.46	0.50	0.51	0.51
1.0	0.17	0.20	0.23	0.25	0.28	0.30	0.33	0.35	0.37	0.40	0.40	0.40
3.0	0.08	0.11	0.14	0.16	0.18	0.20	0.23	0.24	0.25	0.25	0.25	0.27

## Data Availability

Not applicable.

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
