# Peer review of "Influence of Illite and Its Amine Modifications on the Self-Adhesive Properties of Silicone Pressure-Sensitive Adhesives"

_materials, 2023, doi:10.3390/ma16072879_

Round 1
Reviewer 1 Report
The manuscript by Antosik and coworkers describes the use of illite as a filler to improve the thermal properties of a silicone adhesive. The goals are clear, if somewhat limited in scope, and a relatively large number of relevant experiments are presented. However, there are a large number of areas where the level of information and discussion is inadequate.
1. The description of illite is missing some information (lines 60-62). Two-silicon-oxygen tetrahedral joined together do not make an aluminum oxygen octahedron. Further, it would be useful to understand why this particular clay has been tested as a filler.
2. The paragraph starting on line 69 should make it clear that these are literature results and add appropriate references.
3. The main goal is to improve thermal resistance (lines 84-85). However, the text in lines 47-53 indicates that silicone adhesives guarantee stability in a wide temperature range and can be used in devises expose to significant temperature fluctuations. This suggests the properties are already good and may require some clarification.
4. What is N,N-4-amine (line 91 and later)? It appears that the name is not correct especially since it is abbreviated as THA.
5. Table 1 provides details on 3 modified illites. However, there is no information on whether or not the use of higher THA concentration leads to significantly higher amine content in the illite. In fact there is no information at all on the extent of modification.
6. The sentence on lines 120-21 is unclear. Is the pot-life the maximum time after preparation when the material can still be coated on a substrate?
7. Line 144. How is the damage temperature determined? What qualifies as “damage”?
8. Table 2 and Figure 1 suggest that illite and modified illite have virtually identical IR spectra. If that is the interpretation, the text should be clearer on this point.
9. Figure 2 caption should note that Q and I are used to indicate quartz and illite in the XRD plots. The overall intensities here suggest that there is a large amount of quartz? Is that the case?
10. Table 4 presents viscosity results, but the text does not provide information on the range of viscosity that is acceptable or desirable.
11. Tables 4 – 6 present a large amount of data. However, it is not clear whether this represents single data points or whether replicates have been measured. There are, however, errors bars in Figures 3 and 4, indicating that multiple replicates have been measured. Additional information on repeatability of experiments is needed.
12. The discussion of Figure 3 notes the difficulty to interpret the data. This point seems to require clarification of the extent of modification of the samples (how different is the modification for the various THA concentrations) as in point 5 above and the repeatability of experiments as in 11 above.
13. The text on page 9 concludes that the lowest shrinkage values are obtained for neat illite and illite modified with 0.1 M THA. This is not very clear from Table 6 which contains a lot of data; the trend in shrinkage seems to vary considerably for the different conditions (filler content and time) so it would be important to clarify what conditions the statement on shrinkage applies to. The authors might consider if all the data are needed or if the overall trends might be clearer for a subset of the time points. It would also be interesting to add here the shrinkage value for the silicone adhesive in the absence of illite.
14. The Conclusions should provide more context as to how useful the observed change in properties are. For example, does the use of illite actually enable applications that are currently not possible or is it an incremental change that will not have much impact?
English errors
Line 38. “abundant series” does not make sense. Perhaps “major application areas”?
Line 42: dissolving should be dissolution
Line 51: flammability (not flammable)
Line 63: What is mining site? Seems to be an incorrect term.
Line 88: purchased
Line 156. IR band assignments
Line 161: singlet located at 693 cm-1
Line 170: duplicate sentence should be removed
Line 181: while maintaining other parameters…
Line 235: than required
Author Response
The authors would like to thank to the Reviewer and truly appreciate his comments, questions and corrections. Please find the detailed answers below.
Comments and Suggestions for Authors
The manuscript by Antosik and coworkers describes the use of illite as a filler to improve the thermal properties of a silicone adhesive. The goals are clear, if somewhat limited in scope, and a relatively large number of relevant experiments are presented. However, there are a large number of areas where the level of information and discussion is inadequate.
- The description of illite is missing some information (lines 60-62). Two-silicon-oxygen tetrahedral joined together do not make an aluminum oxygen octahedron. Further, it would be useful to understand why this particular clay has been tested as a filler.
We made a change in the manuscript, we hope now it is correct
- The paragraph starting on line 69 should make it clear that these are literature results and add appropriate references.
We made a change in the manuscript, we hope now it is correct
- The main goal is to improve thermal resistance (lines 84-85). However, the text in lines 47-53 indicates that silicone adhesives guarantee stability in a wide temperature range and can be used in devises expose to significant temperature fluctuations. This suggests the properties are already good and may require some clarification.
We made a change in the manuscript, we hope now it is correct
- What is N,N-4-amine (line 91 and later)? It appears that the name is not correct especially since it is abbreviated as THA.
We made a change in the manuscript, we hope now it is correct
- Table 1 provides details on 3 modified illites. However, there is no information on whether or not the use of higher THA concentration leads to significantly higher amine content in the illite. In fact there is no information at all on the extent of modification.
Thank you for your comment, our object of study was not introducing the amine into the filler, we focused on etching the organic layer with the amine. Your comment made us realize that there wasn't enough information about it in the article, so we've added it to the manuscript, we hope now it is correct
- The sentence on lines 120-21 is unclear. Is the pot-life the maximum time after preparation when the material can still be coated on a substrate?
Yes. We made a change in the manuscript, we hope now it is correct
- Line 144. How is the damage temperature determined? What qualifies as “damage”?
We made a change in the manuscript, we hope now it is correct
- Table 2 and Figure 1 suggest that illite and modified illite have virtually identical IR spectra. If that is the interpretation, the text should be clearer on this point.
We made a change in the manuscript, we hope now it is correct
- Figure 2 caption should note that Q and I are used to indicate quartz and illite in the XRD plots. The overall intensities here suggest that there is a large amount of quartz? Is that the case?
We made a change in the manuscript, we hope now it is correct
- Table 4 presents viscosity results, but the text does not provide information on the range of viscosity that is acceptable or desirable.
We made a change in the manuscript, we hope now it is correct
- Tables 4 – 6 present a large amount of data. However, it is not clear whether this represents single data points or whether replicates have been measured. There are, however, errors bars in Figures 3 and 4, indicating that multiple replicates have been measured. Additional information on repeatability of experiments is needed.
Thank you for pointing it out, it was missing in the text. We made a change in the manuscript, we hope now it is correct
- The discussion of Figure 3 notes the difficulty to interpret the data. This point seems to require clarification of the extent of modification of the samples (how different is the modification for the various THA concentrations) as in point 5 above and the repeatability of experiments as in 11 above.
We made a change in the manuscript, we hope now it is correct
- The text on page 9 concludes that the lowest shrinkage values are obtained for neat illite and illite modified with 0.1 M THA. This is not very clear from Table 6 which contains a lot of data; the trend in shrinkage seems to vary considerably for the different conditions (filler content and time) so it would be important to clarify what conditions the statement on shrinkage applies to. The authors might consider if all the data are needed or if the overall trends might be clearer for a subset of the time points. It would also be interesting to add here the shrinkage value for the silicone adhesive in the absence of illite.
We made a change in the manuscript, we hope now it is correct
- The Conclusions should provide more context as to how useful the observed change in properties are. For example, does the use of illite actually enable applications that are currently not possible or is it an incremental change that will not have much impact?
We made a change in the manuscript, we hope now it is correct
English errors
Line 38. “abundant series” does not make sense. Perhaps “major application areas”?
Line 42: dissolving should be dissolution
Line 51: flammability (not flammable)
Line 63: What is mining site? Seems to be an incorrect term.
Line 88: purchased
Line 156. IR band assignments
Line 161: singlet located at 693 cm-1
Line 170: duplicate sentence should be removed
Line 181: while maintaining other parameters…
Line 235: than required
We made a change in the manuscript, in addition, we checked the manuscript for language, we hope now it is correct
Finally, we hope that corrections made in the manuscript fulfill reviewer suggestions and allow editor to make positive decision about acceptation of our contribution for publishing in this journal.
With regards,
Adrian Krzysztof Antosik
Karolina Mozelewska
Marlena Musik
Piotr MiÄ…dlicki
Katarzyna Wilpiszewska
Reviewer 2 Report
The manuscript is presented in a structured manner, documented, and informative. The introduction provides sufficient background and includes relevant references. The study is correctly designed and technically sound. The results are enough to draw consistent conclusions with the evidence and arguments presented in the article.
Recommendations for Authors
1. References 1 and 8 are the same, please rectify them.
2. In the paragraph from the first page lines 31 – 39 at the end the authors have references 5-7. References 6 and 7 should be replaced with ones related to the text.
*reference 6 (García-Segura, A.; Sutter, F…… doi:10.1016/j.rser.2021.110879.) should not be put here, considering that in this reference it is about the identification of the possible degradation of solar reflectors and a survey of the durability tests most commonly used, as well as the main degradation types reported in the literature for different reflectors materials. I do not see the importance of this paper for the self-adhesives are widely used….
*reference 7 (van der Borst, M.; ….. doi:10.1016/S0951-8320(01)00007-2.) is about Probabilistic safety assessment (PSA) of a nuclear power plant not about “products meet high European standards, and the worldwide guarantee of the desired functions and properties of the products is of particular importance, especially in the case of abundant series, like in the automotive industry”
*also check reference 7 from second page line 59.
3. Page 2 line 70 “(N2H4)” and line 71 “N2H4”, please choose with or without subscript.
*the authors should decide if they use superscript or not, because on line 109 they have “cm-1” and from line 156 they used “cm-1”.
4. Page 6 line 191 the authors wrote “…filler modified with sulfuric acid…”, but in the 2.2. Filler modification section doesn’t write anything about the sulfuric acid. Please be specific.
5. Page 7 line 206 “….and is common in the technology of [18].”, the authors should not finish the sentence with a reference, they could explain in a few words what is about in reference 18.
6. Page 7 the paragraph from line 206 – 210 “However, it is 206 difficult to interpret……This is quite a rare phenomenon that is difficult to interpret.”
I understand that it is difficult to interpret, but still, the authors gave an interpretation so only mentioning once will be sufficient. The authors should make modifications to the above mentioned lines.
7. Conclusions should be numbered with 4 not 5.
8. Table 5 should have the symbols of modified fillers (THA 0….) presented on columns the same as tables 4 and 6.

Author Response
The authors would like to thank to the Reviewer and truly appreciate his comments, questions and corrections. Please find the detailed answers below.
The manuscript is presented in a structured manner, documented, and informative. The introduction provides sufficient background and includes relevant references. The study is correctly designed and technically sound. The results are enough to draw consistent conclusions with the evidence and arguments presented in the article.
Recommendations for Authors
- References 1 and 8 are the same, please rectify them.
We made a change in the manuscript, we hope now it is correct
- In the paragraph from the first page lines 31 – 39 at the end the authors have references 5-7. References 6 and 7 should be replaced with ones related to the text.
*reference 6 (García-Segura, A.; Sutter, F…… doi:10.1016/j.rser.2021.110879.) should not be put here, considering that in this reference it is about the identification of the possible degradation of solar reflectors and a survey of the durability tests most commonly used, as well as the main degradation types reported in the literature for different reflectors materials. I do not see the importance of this paper for the self-adhesives are widely used….
*reference 7 (van der Borst, M.; ….. doi:10.1016/S0951-8320(01)00007-2.) is about Probabilistic safety assessment (PSA) of a nuclear power plant not about “products meet high European standards, and the worldwide guarantee of the desired functions and properties of the products is of particular importance, especially in the case of abundant series, like in the automotive industry”
*also check reference 7 from second page line 59.
We made a change in the manuscript, we hope now it is correct
- Page 2 line 70 “(N2H4)” and line 71 “N2H4”, please choose with or without subscript.
*the authors should decide if they use superscript or not, because on line 109 they have “cm-1” and from line 156 they used “cm-1”.
We made a change in the manuscript, we hope now it is correct
- Page 6 line 191 the authors wrote “…filler modified with sulfuric acid…”, but in the 2.2. Filler modification section doesn’t write anything about the sulfuric acid. Please be specific.
We made a change in the manuscript, we hope now it is correct
- Page 7 line 206 “….and is common in the technology of [18].”, the authors should not finish the sentence with a reference, they could explain in a few words what is about in reference 18.
We made a change in the manuscript, we hope now it is correct
- Page 7 the paragraph from line 206 – 210 “However, it is 206 difficult to interpret……This is quite a rare phenomenon that is difficult to interpret.”
I understand that it is difficult to interpret, but still, the authors gave an interpretation so only mentioning once will be sufficient. The authors should make modifications to the above mentioned lines.
We made a change in the manuscript, we hope now it is correct
- Conclusions should be numbered with 4 not 5.
We made a change in the manuscript, we hope now it is correct
- Table 5 should have the symbols of modified fillers (THA 0….) presented on columns the same as tables 4 and 6.
In table 5, the symbols of modified fillers are the same as in tables 4 and 6. Please note that the first column shows the filling quantity of the adhesive film with the filler. Filler symbols are at the top of columns 2-5.
Finally, we hope that corrections made in the manuscript fulfill reviewer suggestions and allow editor to make positive decision about acceptation of our contribution for publishing in this journal.
With regards,
Adrian Krzysztof Antosik
Karolina Mozelewska
Marlena Musik
Piotr MiÄ…dlicki
Katarzyna Wilpiszewska
Round 2
Reviewer 1 Report
The authors have resolved most of the questions raised in my previous review. However, there is still a lack of clarity about the composition of the illite used, before and after etching with THA.
It is now clear that the use of THA is to remove the organic content from the illite by an etching process, which was not clear in the original manuscript. However, section 2.1 (Materials) simply refers to illite and does not specify what the organic content is. Furthermore, my original question on the extent of “modification” by THA is still relevant, but should be rephrased. What is the residual organic filler content for each of the 3 THA concentrations used—is it significantly different and are the differences responsible for some of the variation in properties that is observed.
The paper is acceptable for publication after the above question has been dealt with.
Author Response
The authors would like to thank to the Reviewer and truly appreciate his comments, questions and corrections. Please find the detailed answers below.
The authors have resolved most of the questions raised in my previous review. However, there is still a lack of clarity about the composition of the illite used, before and after etching with THA.
It is now clear that the use of THA is to remove the organic content from the illite by an etching process, which was not clear in the original manuscript. However, section 2.1 (Materials) simply refers to illite and does not specify what the organic content is. Furthermore, my original question on the extent of “modification” by THA is still relevant, but should be rephrased. What is the residual organic filler content for each of the 3 THA concentrations used—is it significantly different and are the differences responsible for some of the variation in properties that is observed.
We made a change in the manuscript, we hope now it is correct
The paper is acceptable for publication after the above question has been dealt with.
Finally, we hope that corrections made in the manuscript fulfill reviewer suggestions and allow editor to make positive decision about acceptation of our contribution for publishing in this journal.
With regards,
Adrian Krzysztof Antosik
Karolina Mozelewska
Marlena Musik
Piotr MiÄ…dlicki
Katarzyna Wilpiszewska